# Contrast-enhanced and indirect computed tomography lymphangiography accurately identifies the cervical lymphocenter at risk for metastasis in pet dogs with spontaneously occurring oral neoplasia

Stephanie Goldschmidt[1]*, Nikia Stewart[2], Christopher Ober[1], Cynthia Bell[3], Amber Wolf-Ringwall[1], Michael Kent[4], Jessica Lawrence[1]

1 Department of Veterinary Clinical Sciences, College of Veterinary Medicine, University of Minnesota, St Paul, Minnesota, United States of America, 2 VETCT, Orlando, Florida, United States of America, 3 Specialty Oral pathology for Animals, Geneseo, Illinois, United States of America, 4 Department of Surgical and Radiologic Sciences, School of Veterinary Medicine, University of California Davis, Davis, California, United States of America

* sgoldschmidt@ucdavis.edu

## Abstract

For dogs with oral tumors, cervical lymph node (LN) metastasis alters treatment and prognosis. It is therefore prudent to make an accurate determination of the clinical presence (cN+ neck) or absence (cN0 neck) of metastasis prior to treatment. Currently, surgical LN extirpation with histopathology is the gold standard for a diagnosis of metastasis. Yet, recommendations to perform elective neck dissection (END) for staging are rare due to morbidity. Sentinel lymph node (SLN) mapping with indirect computed tomography lymphangiography (ICTL) followed by targeted biopsy (SLNB) is an alternative option to END. In this prospective study, SLN mapping followed by bilateral END of all mandibular LNs (MLNs) and medial retropharyngeal LNs (MRLNs) was performed in 39 dogs with spontaneously occurring oral neoplasia. A SLN was identified by ICTL in 38 (97%) dogs. Lymphatic drainage patterns were variable although most often the SLN was identified as a single ipsilateral MLN. In the 13 dogs (33%) with histopathologically confirmed LN metastasis, ICTL correctly identified the draining lymphocentrum in all (100%). Metastasis was confined to the SLN in 11 dogs (85%); 2 dogs (15%) had metastasis beyond the SLN ipsilaterally. Contrast enhanced CT features had good accuracy in predicting metastasis, with short axis measurements less than 10.5 mm most predictive. ICTL imaging features alone were unable to predict metastasis. Cytologic or histopathologic SLN sampling is recommended prior to treatment to inform clinical decision-making. This is the largest study to show potential clinical utility of minimally invasive ICTL for cervical LN evaluation in canine oral tumors.

**Data Availability Statement:** All relevant data are within the paper and its Supporting information files.

**Funding:** University of Minnesota Grant in Aid PI: SG, CO-I: JL,CO,NS Grant #: 1801 - 11652 - 20562 - 4214572 https://research.umn.edu/funding-awards/grant-aid The funders had no role in study design, data collection and analysis, decision to publish, or preparation of the manuscript.

**Competing interests:** The authors have declared that no competing interests exist.

# Introduction

Cervical lymph node (LN) metastasis is a negative prognostic factor for oral malignant melanoma (OMM), mast cell tumor (MCT), and oral squamous cell carcinoma (OSCC) [1–9]. Yet, there is currently a lack of consensus among specialists for LN staging methodologies in canine oral tumors [10]. The gold standard for diagnosis of LN metastasis remains histopathologic diagnosis; however, palpation, computed tomography (CT) features, and fine needle aspiration cytology are commonly utilized to provide an assessment of the clinical presence (cN+ neck) or absence (cN0 neck) of metastasis [10]. In dogs with OMM, LN palpation is inaccurate in detecting metastatic disease with up to 40% of LNs deemed normal on palpation harboring metastasis [11]. CT allows evaluation of the shape and size of the cervical LN to predict metastasis, but sensitivity varies widely from 12–83% [12,13]. FDG-PET-CT does not improve detection of metastasis compared to contrast-enhanced CT alone [13]. Fine needle aspiration cytology has moderate accuracy for canine cervical LNs, with one study reporting a false negative rate of 8.1% for head and neck tumors [14]. Specific neoplastic types present additional challenges for cytologic detection, with reported false negative rates of up to 36% in dogs with OMM and oral fibrosarcoma compared to histopathological assessment [3,15,16]. The lack of consistent detection of cervical LN metastasis prior to treatment limits the ability to quantify the absolute impact of metastasis on progression-free survival.

Furthermore, canine oral lymphatic drainage is unpredictable and up to 62% of oral tumors have contralateral dissemination [17]. Clinicians often sample ipsilateral and contralateral LNs in dogs with oral tumors. However, only the lateral mandibular LNs (MLNs) or grossly abnormal LNs on palpation are often sampled due to their superficial, accessible location. Two histopathologic studies evaluated all 3 lymphocentrums ipsilateral to head and neck neoplasia and found that 26.7–45.5% of neoplasms spread to LNs other than the MLNs [14,18]. In another study of OSCC and OMM, 6% of dogs had medial retropharyngeal LN (MRLN) metastasis without concurrent spread to an ipsilateral MLN [19]. This highlights the potential for incomplete staging if only the lateral MLN is sampled, which could affect surgical or radiotherapy approaches.

Concerns surrounding missing occult cervical metastasis have led to recommendations for pathologic staging (pN+ versus pN0 neck) of all, or a subset of, cervical LNs with elective neck dissection (END) [20–22]. END carries a risk of surgical morbidity including post-operative edema, seroma formation and infection. While major complications are rare [10,20], there is no known survival benefit following removal of normal LNs. Accordingly, END is rarely used in the context of normal appearing LN (cN0 neck) on diagnostic imaging [10]. In a recent survey, veterinary specialists responded that they recommend END in the cN0 neck in 13–28% of dogs with T1-T3 OSCC, 25–38% of dogs with T1-T3 OMM, and 27% of dogs with oral MCT [10]. Bilateral MLN and MRLN extirpation is most commonly recommended for canine OMM and oral MCT; bilateral or ipsilateral MLN and MRLN removal is recommended with equal frequency for OSCC [10].

Sentinel LN (SLN) mapping and biopsy for pathologic staging is an alternative to END. The SLN is the first LN that drains metastatic deposits from the primary tumor. SLN mapping and biopsy (SLNB) is standard practice for several histologic types in humans because studies have shown that if the SLN is negative for metastasis, then additional draining LNs are negative in over 90% of patients [23–25]. Validation studies for SLN mapping for human oral carcinoma have reported low FN rates ranging from 2–14% [26–28]. Comparisons of SLNB to END in the cN0 neck report similar regional recurrence and 5-year disease-free survival [26,29]. Importantly, due to negative SLNB, approximately 70% of patients were spared from undergoing END [29]. Because of its clinical utility, SLNB as listed as an alternative to END for early

stage (T1 or T2) oral cavity carcinoma for identification of metastasis in the National Comprehensive Cancer Network guidelines [30].

The use of SLN mapping has been evaluated in small cohorts of veterinary patients [31–35]. Indirect CT lymphangiography (ICTL) specifically has been evaluated in canine head and neck tumors, with a draining LN identified in 55–89% of dogs [31,36–39]. The majority of previous studies did not pathologically confirm that the SLN on imaging accurately predicted the remainder of the cervical lymphatic basin. This precludes the confident incorporation of ICTL into standard practice. The primary purpose of this study was to determine if ICTL with SNLB could accurately predict cervical LN metastasis (pN+ neck) in dogs with spontaneously occurring oral neoplasia. A corresponding aim was to compare the diagnostic accuracy of imaging features from ICTL to the standard imaging approach, post-contrast CT, for prediction of metastasis to the SLN. Here, we demonstrate that ICTL reliably identifies the draining cervical LN basin at risk for metastasis in common canine oral tumors and that it may provide complementary information to post-contrast CT image evaluation. We also confirm previous findings that the diagnostic accuracy of imaging features is insufficient to diagnose cervical metastasis.

## Materials and methods

### Dogs

Client owned dogs with spontaneously occurring oral and perioral neoplasia were prospectively enrolled from May 2019-May 2022. Inclusion criteria included dogs with any oral pathology that had confirmed or highly suspicious cervical metastasis (cN+ neck) based on palpation, diagnostic imaging, or cytology. Dogs with the cN0 neck were recruited if the primary oral pathology was associated with a ≥20% risk of locoregional spread. Specifically, based on current literature, this included T1-T3 OMM [3,4,8,19,40–42], T2-T3 OSCC [2,19,43,44], and T1-T3 oral/perioral MCT [1,9]. Furthermore, dogs with a primary oral tumor and distant metastasis were deemed at high risk for LN metastasis and were also eligible for enrollment. Both grossly visible masses and excisional biopsy scars were permitted if they met the other inclusion criteria. Dogs were excluded if prior cervical dissection or locoregional radiation therapy had occurred, or if there was a contra-indication to intravenous contrast administration.

Tumor size was categorized based on the World Health Organization grading scheme [45]. Prior chemotherapy, radiation therapy or immunotherapy were not permitted. The study was approved by the Institutional Animal Care and Use Committee (IACUC #2201-39790A). Written client consent was obtained prior to enrollment. Clinical data, including signalment, weight, tumor histology, tumor longest diameter, and tumor location were recorded. In dogs presenting with microscopic disease, the original documented tumor diameter was recorded. Tumor location was categorized by laterality and as rostral maxilla, caudal maxilla, rostral mandible, caudal mandible, mucosal (not directly overlying bone), or tongue. Caudal was defined as distal to the second premolar [46].

### Image acquisition

All dogs were anesthetized through the anesthesia service overseen by a board-certified anesthesiologist. All dogs were positioned for CT in sternal recumbency, with the head positioned on a 3D printed bite block (Fig 1) to elevate the oral cavity and limit pressure on draining lymphatics. CT images were acquired using a helical 64-slice scanner (Aquilion 64 CFX, Toshiba Medical Systems, Tustin, CA with maximum field of view of 70 cm) with the following parameters suitable to dog size: 120kVp, 40-80mA, 20-100mAs, rotation time 0.5 seconds. Intravenous contrast medium (770 mg of Iodine/kg [2 mL/kg]; Optiray 350 [Ioversol], Mallinckrodt

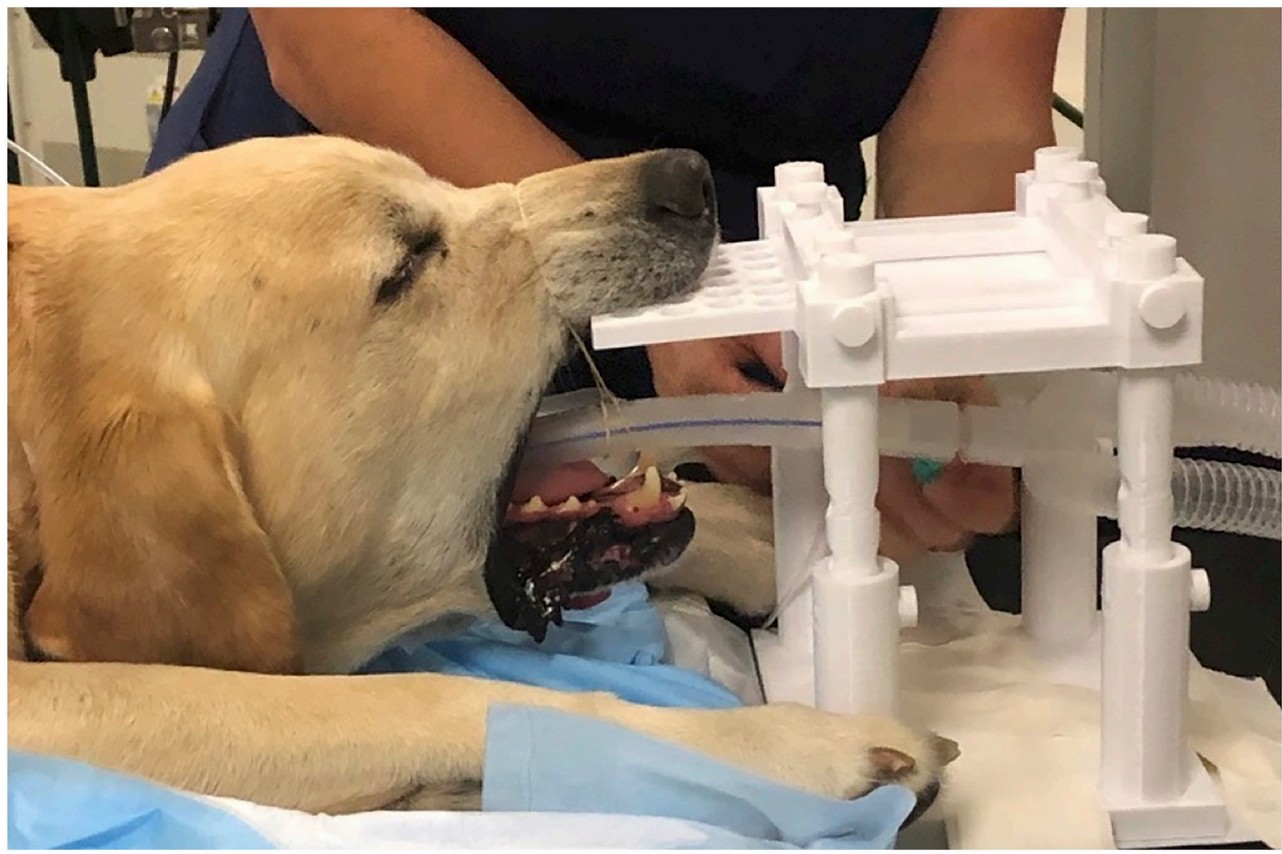

**Fig 1. Positioning for indirect computed tomography lymphangiography (ICTL) procedure.** A dog with an oral tumor positioned in sternal recumbency with the rostral maxilla positioned on a 3-D printed bite block in preparation for computed tomography (CT) scan. All dogs were positioned similarly to limit compression on the ventral neck for improve cervical drainage. CT images were subsequently acquired with pre- and post-contrast images followed by additional CT image acquisition 3, 6 +/-12 minutes (min) following peritumoral contrast injection.

Inc, Hazelwood, MO) was administered intravenously to all dogs following acquisition of pre-contrast data. All examinations consisted of reconstructed 2- or 3-mm thick transverse images and datasets were stored to allow analysis.

After post-contrast scan, ICTL was performed as previously described [36,37,39] with minor modifications. Briefly, 1 ml of Ioversol was diluted with 1 ml of saline. Four site peritumoral injections (0.5 ml per site) were performed with a 23-gauge butterfly catheter. Each injection was performed over 30 seconds followed by massage of the region. Scans were acquired 3 minutes (min) and 6 min from the start of the peritumoral injection. If no SLN was noted on the 6-min scan, an additional 1 ml of undiluted Ioversol was injected peritumorally in the same fashion. A repeat scan was performed 3 min later, at 12 min from the initial injection. Complications associated with the ICTL procedure were recorded.

## Image analysis

The post-contrast CT and ICTL images were evaluated by 2 board-certified radiologists (NS, CO) and a board-certified radiation oncologist (JL). The first ICTL study (3, 6-, or 12-minute scan) where a SLN(s) could be identified was used for evaluation. Using a standardized scoring form created in a commercially available software program (Excel v.16, Microsoft Corporation, Redmond WA), each investigator evaluated six distinct LNs independently: the right and

**Table 1. Cervical LN classification for dogs with oral tumors on post-contrast CT.**

| CT Characteristic | Score | Criteria |
|---|---|---|
| LN Size | | Maximum long axis diameter: |
| | 0 | 0-5mm |
| | 1 | 5-10mm |
| | 2 | >10mm |
| LN Shape | 0 | Normal |
| | 1 | Irregular |
| Contrast Pattern | 0 | Absent/homogenous opacification |
| | 1 | Heterogeneous opacification |
| Subjective Classification Score | 0 | Non-metastatic |
| | 1 | Possibly metastatic |
| | 2 | Metastatic |
| Composite CT Score A | 0–4 | Sum of size, shape & contrast pattern scores |
| Composite CT Score B | 0–6 | Sum of size, shape, contrast pattern, & subjective scores |

left medial and lateral MLN and the right and left MRLN. All evaluators were blinded to histopathologic status of the LN at the time of review and to each other's assessment.

## Post-contrast CT analysis

A standardized, previously published grading scheme for LN assessment [13] was used with minor modifications. LN size was measured in mm by its long axis, defined as the maximum diameter in the transverse plane, and its short axis, defined as the diameter perpendicular to the long axis. The mean axis measurements from the 3 reviewers were calculated. Each reviewer used a standardized scoring rubric to classify each cervical LN (Table 1). Subjective scoring was was based on the lymph node size, shape, and contrast pattern, and dependent on clinician experience. When there was not consensus between all reviewers, the majority (2/3 reviewers) score was used for analysis.

## ICTL analysis

The identified SLN(s) were recorded. A standardized grading scheme was applied based on a combination of previously published grading schemes (Table 2) [13,39]. Subjective grading was not performed. Mean Hounsfield Units (HU) were measured in the center and periphery of the SLN by a single reviewer (NS), using a circular region of interest 1.0mm in diameter. The region of interest for SLN periphery was defined as 1.0mm from the LN capsule. Two measurements were taken at each location from opposing quadrants and averaged. To evaluate change in HU from the post-contrast images, mean central and peripheral HU were measured

**Table 2. Criteria used to classify cervical LNs from dogs with oral tumors from ICTL images.**

| CT Characteristic | Score | Criteria |
|---|---|---|
| ICTL Contrast Pattern | 0 | Absent/homogenous opacification |
| | 1 | Heterogeneous opacification |
| | 2 | Peripheral opacification |
| Composite ICTL Score | 0–5 | Sum of size[a], shape[a] & ICTL contrast pattern scores |

[a] Determined from post-contrast CT analysis (Table 1).

on the identified SLN on the post-contrast CT. When additional MLN past the medial and lateral were identified as SLN(s) these were not evaluated.

## Surgery and histopathologic assessment

Within 2 weeks of CT scan, surgical removal of MLNs and MRLNs was performed bilaterally. Surgery was performed by a board-certified dentist and oral surgeon (SG) or board-certified oncologic surgeon (PA). Surgery was performed through two lateral incisions or a single ventral incision depending on surgeon preference. Intra- and post-operative complications were recorded.

Resected LNs were submitted for histopathological evaluation by a single board-certified pathologist (CB). Sectioning of LNs was performed as serial 2.5mm cross sections perpendicular to the long axis. All resulting pieces of tissue were processed, embedded, stained with hematoxylin and eosin, and examined histologically. Exceptions were made for LNs that were grossly enlarged, abnormal, and considered likely metastatic. For these LNs, representative sections were sampled to confirm metastasis, identify the tissue as LN origin, and determine presence or absence of extra-nodal extension. Histopathology results were reported as micro ($< 2$mm) or macro ($>2$ mm) metastasis.

## Statistical analysis

Statistical analyses were conducted using a commercially available statistics program (Stata version 14.2, Stata Corporation, College Station, Texas, USA). Descriptive statistics were performed. Continuous data was assessed for normality by visualization of distributional plots and use of a Shapiro-Wilk normality test. When continuous data was normally distributed, means and standard deviations were reported; otherwise, medians and overall range were reported. Totals and percentages were used to describe categorical data. Either a Chi square or Fisher's exact test was used to look for differences in proportions between categorical data. Logistic regression was done to look for effects of different evaluated scales for the risk of metastasis in each lymph node or lymph node grouping. Receiver operator curves were then graphed for sensitivity versus 1-sensitivity and an area under the curve calculated. This analysis for the accurate diagnosis of metastasis (micro or macro-metastasis) was performed for LN size (short axis, long axis), shape score, contrast score, CT score A, CT score B, ICTL contrast score, ICTL composite score, HU values for SLN, and change in HU values between post-contrast and ICTL. Inter-observer agreement was assessed for the CT and ICTL scores on each LN separately using Kappa-weighted statistics. Kappa scores were interpreted as: slight (0.0–0.20), poor (0.21–0.40), moderate (0.41–0.60), substantial (0.61–0.79), excellent (0.81–1.0) [47]. For size measurements, intraclass correlation coefficient (ICC) were calculated and interpreted as $<0.5$ poor, 0.5–0.75 moderate, 0.75–0.90 good, and $>0.90$ excellent [48]. P values $< 0.05$ were considered significant.

## Results

### Clinical characteristics

Thirty-nine dogs were enrolled in the study (S1 Table). OMM was the predominant tumor type (Table 3). There were variable anatomic locations and relatively even tumor size distribution (Table 4). At SLN mapping, 22 dogs (56%) had gross tumor and 17 (44%) had scars. In these latter 17 dogs, the median (range) time from excisional biopsy to SLN mapping was 26 (16–117) days.

**Table 3. Histopathologic diagnosis of tumor types and presence or absence of LN metastasis in dogs with oral tumors.**

| Final histopathologic diagnosis (N = 39) | # affected dogs (%) | Discordant initial biopsy (reported below) and final histo-pathologic diagnosis | Clinical note |
|---|---|---|---|
| **Oral malignant melanoma (OMM)** | | | |
| # of dogs (%) | 26 (67%) | | |
| # LN metastasis (%) | 7 (28%) | | |
| **Oral squamous cell carcinoma (OSCC)** | | | |
| # of dogs (%) | 4 (10%) | | |
| # LN metastasis (%) | 2 (50%) | | |
| **Mast cell tumor (MCT)** | | | |
| # of dogs (%) | 2 (5%) | | |
| # LN metastasis (%) | 2 (100%) | | |
| **Amyloid producing odontogenic tumor (APOT)** | | OSCC (N = 1) | Pulmonary metastasis (N = 1)[a] |
| # of dogs (%) | 2 (5%) | | |
| # LN metastasis (%) | 0 (0%) | | |
| **Ameloblastoma** | | OSCC | |
| # of dogs (%) | 1 (2.6) | | |
| # LN metastasis (%) | 0 (0%) | | |
| **Carcinoma in Situ** | | OSCC | |
| # of dogs (%) | 1 (2.6%) | | |
| # LN metastasis (%) | 0 (0%) | | |
| **Fibrosarcoma** | | | Highly suspicious cervical metastasis (cN+ neck) on palpation |
| # of dogs (%) | 1 (2.6%) | | |
| # LN metastasis (%) | 1 (100%) | | |
| **Hemangiosarcoma** | | | Non-diagnostic biopsy, OSCC favored based on CT imaging features |
| # of dogs (%) | 1 (2.6%) | | |
| # LN metastasis (%) | 0 (0%) | | |
| **Epitheliotropic Lymphoma** | | MCT | |
| # of dogs (%) | 1 (2.6%) | | |
| # LN metastasis (%) | 1 (100%) | | |

[a]Pulmonary metastasis in one dog's APOT was reported elsewhere [49].

Mucosal tumors occurred in 16 dogs (41%) while 15 dogs (38%) had unilateral tumors overlying/involving bone, 5 dogs (13%) had bilateral tumors overlying/involving bone, and 3 dogs (8%) had lingual tumors (Table 5). OMM was significantly (p = 0.001) more common in the mucosal location than other tumor types (Table 3).

LN metastasis was confirmed in 13 dogs (33%) and in 22 of 234 LN examined (9%). Of the 22 metastatic lymph nodes, 17 (77%) were characterized by macrometastasis. OMM was the most common histology associated with metastasis (Table 3) but this was not significant (p = 0.16). Likewise, tumor location was not significantly (p = 0.53) associated with LN metastasis.

## Interobserver agreement for post-contrast CT and ICTL image features

For most imaging features, the agreement between reviewers was moderate with the shape, ICTL contrast pattern, and composite ICTL scores more likely to have substantial agreement

**Table 4. Tumor size distribution in dogs with oral tumors.**

| Category | Number (%) |
|---|---|
| **T-stage all (N = 39)** | |
| 1 (<2 cm) | 14 (36%) |
| 2 (2–4 cm) | 11 (28%) |
| 3 (>4 cm) | 14 (36%) |
| **T-stage of tumors grossly present (N = 22)** | |
| 1 (<2 cm) | 2 (9%) |
| 2 (2–4 cm) | 7 (32%) |
| 3 (>4 cm) | 13 (59%) |
| **T-stage determined from medical record when dogs presented with surgical scar (= 17)** | |
| 1 (<2 cm) | 12 (71%) |
| 2 (2–4 cm) | 4 (24%) |
| 3 (>4 cm) | 1 (6%) |

**Table 5. Primary tumor location in 39 dogs with oral tumors and identified sentinel lymph nodes (SLNs) with ICTL for prediction of metastasis to the right and/or left mandibular lymph nodes (MLNs), and/or medial retropharyngeal lymph node (MLN).**

| Location (N = 39 dogs) | SLN(s) (N = 38 dogs*) | | | | | |
|---|---|---|---|---|---|---|
| | Ipsilateral MLN(s) | Ipsilateral MLN(s) + MRLN | Ipsilateral MRLN | Contra-lateral MLN(s) | Contra-lateral MLN (s) + MRLN | Ipsilateral and Contra-lateral |
| **Mucosal (N = 16)** | **11/16 (69%)** | **4/16 (25%)** | - | - | - | **1/16 (6%)** |
| Buccal mucosa of the lip (n = 5) | 4/5 (80%) | 1/5 (10%) | - | - | - | - |
| Buccal mucosa in the cheek pouch (n = 4) | 3/4 (75%) | 1/4 (25%) | - | - | - | - |
| Labial Frenulum (n = 4) | 2/4 (50%) | 1/4 (25%) | - | | | 1/4 (25%) ipsilateral MLN + MRLN & contralateral MRLN |
| Mucocutaneous junction (n = 3) | 2/3 (67%) | 1/3 (33%) | - | | | |
| **Unilateral tumors overlying bone (N = 15)** | **13/14 (93%)** | **1/14 (7%)** | **-** | **-** | **-** | **-** |
| Caudal Mandible (n = 7) | 6/7 (86%) | 1/7 (14%) | - | - | - | - |
| Caudal maxilla (n = 4) | 4/4 (100%) | - | - | - | - | - |
| Rostral maxilla (n = 2) | 2/2 (100%) | - | - | - | - | - |
| Rostral mandible (n = 2)* | 1/1 (100%) | - | - | - | - | - |
| **Bilateral tumors overlying bone (N = 5)** | **2/5 (40%)** | **-** | **-** | **-** | **-** | **3/5 (60%)** |
| Rostral bilateral mandible (n = 4) | 2/4 (50%) | - | - | - | - | 2/4 (50%)—MLN bilaterally |
| Rostral bilateral maxilla (n = 1) | - | - | - | - | - | 100% (1/1)—MLN bilaterally |
| **Tongue (N = 3)** | **-** | **-** | **1/3 (33%)** | | | **2/3 (67%) MRLN bilaterally** |

*note that 1 dog with a unilateral rostral mandibular tumor did not have an SLN identified.

**Table 6. Kappa agreement between reviewers for computed tomography (CT) and indirect computed tomography lymphangiography (ICTL) image feature scores.**

| Imaging Feature | Right medial MLN | Right lateral MLN | Right MRLN | Left medial MLN | Left lateral MLN | Left MRLN |
|---|---|---|---|---|---|---|
| Shape Score | 0.76 | 0.49 | 0.3 | 0.61 | 0.66 | 0.65 |
| Contrast Score | 0.71 | 0.78 | 0.49 | 0.52 | 0.58 | 0.41 |
| Subjective Score | 0.71 | 0.56 | 0.43 | 0.53 | 0.53 | 0.51 |
| CT score A | 0.58 | 0.43 | 0.44 | 0.52 | 0.51 | 0.44 |
| CT Score B | 0.5 | 0.45 | 0.42 | 0.42 | 0.49 | 0.39 |
| ICTL Contrast Score | 0.72 | 0.73 | 0.51 | 0.78 | 0.75 | 0.56 |
| Composite ICTL Score | 0.66 | 0.62 | 0.53 | 0.68 | 0.63 | 0.56 |

Kappa agreement was color coded for poor (white), moderate (light grey), and substantial (dark grey). This was done for the mandibular lymph nodes (MLN) and medial retropharyngeal lymph nodes (MRLN).

(Table 6). Short and long axis measurements of each LN varied minimally between reviewers. ICC agreement was excellent (0.92–0.98) for all LN sizes, except the left MRLN short axis which was good (0.81) and the right medial MLN short and long axis which were both poor (0.29 and 0.39, respectively). Differences in median measurements for each of the 6 nodes varied between reviewers from 0.1–0.7mm for the short axis and 0.2–1.0 mm for the long axis.

## Post-contrast CT imaging analysis

The post-contrast CT features were assessed separately for each of the 6 distinct LN (left and right medial MLN, lateral MLN and MRLN), and then combined into categories including MLN (N = 156), MRLN (N = 78) and all LN (N = 234). Combined data is shown below, additional data on each distinct LN category is available in S2 Table. All imaging features had acceptable to good diagnostic accuracy for prediction of metastasis (Table 7). Increasing scores for all features were significantly associated with metastasis.

**Table 7. Significance of logistic regression model and receiver operator characteristic (ROC) curve analysis for post-contrast computed tomography feature prediction of lymph node (LN) metastasis.**

| Characteristic | | All LN (N = 234) | MLN (N = 156) | MRLN (N = 78) |
|---|---|---|---|---|
| LN metastasis | | | | |
| | Number | 22 | 16 | 4 |
| | Percentage | 9.4% | 11.5% | 5.1% |
| Short axis length | | AUC: 0.83 $P<0.0001$ | AUC: 0.83 $P<0.0001$ | AUC: 0.90 $P<0.0001$ |
| Long axis length | | AUC: 0.74 $P<0.0001$ | AUC: 0.77 $P<0.0001$ | AUC: 0.77 $P<0.02$ |
| Shape score | | AUC: 0.72 $P<0.0001$ | AUC: 0.72 $P = 0.0002$ | AUC: 0.66 $P = 0.15$ |
| Contrast score | | AUC: 0.70 $P = 0.0002$ | AUC: 0.69 $P = 0.002$ | AUC: 0.74 $P = 0.05$ |
| Subjective score | | AUC: 0.77 $p<0.0001$ | AUC: 0.76 $P<0.0001$ | AUC: 0.82 $P = 0.0007$ |
| CT score A | | AUC: 0.75 $P<0.0001$ | AUC: 0.75 p $P = 0.0001$ | AUC: 0.79 $P<0.0001$ |
| CT score B | | AUC:0.79 $P<0.0001$ | AUC: 0.79 $P<0.0001$ | AUC: 0.82 $P = 0.007$ |

Data for all lymph nodes, mandibular lymph nodes (MLN) alone, and medial retropharyngeal lymph nodes (MRLN) alone are shown with corresponding P-values and area under the curves (AUC). Significant values are shaded light grey.

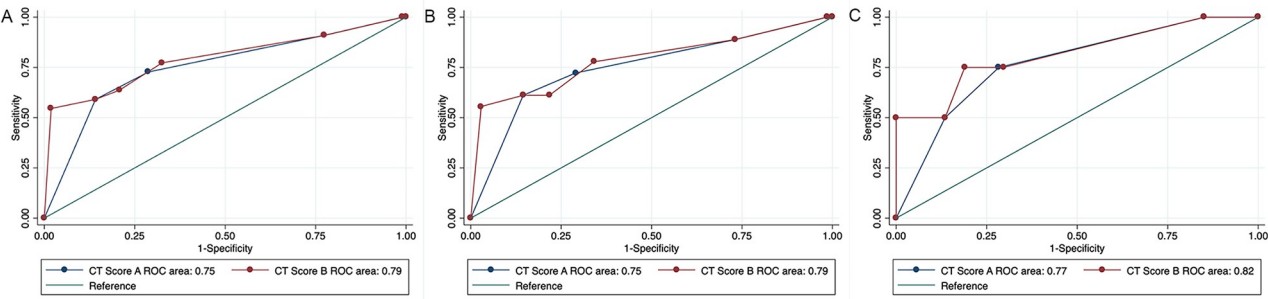

**Fig 2. Receiver operator characteristic (ROC) curves illustrate the diagnostic ability of CT composite scores for the prediction of cervical lymph node metastasis in dogs with oral tumors.** CT score A, consisting of lymph node (LN) size, shape and contrast pattern scores, was inferior to CT score B, which included CT Score A features with a subjective metastasis score. CT score B was significantly superior for the prediction of metastasis when all LN were categorized together (**A**; P = 0.0005)) and when MLN (**B**; P = 0.05) or MRLN (**C**; P = 0.03) were evaluated separately.

CT score B had superior diagnostic accuracy (P = 0.0005) compared to CT score A (Fig 2). There was overlap in scores for metastatic and non-metastatic LN, although this was minimal with short axis measurements (Figs 3 and 4). Indeed, short axis measurements performed best in accurately diagnosing metastasis (Table 7). The median (range) short axis of all cervical LNs was 5.3 (2.5–27.9) mm. The maximum short axis of LNs without metastasis was 10.5 mm. The maximum short axis or MLN and MRLN without metastasis were 10.5 and 10.4 mm, respectively (Fig 4). Conversely, the median (range) long axis of all cervical LNs was 12.13 mm (4.23–36.7 mm). The maximum long axis for LN without metastasis was 33.6 mm and the maximum long axis measurement of 36.7 mm for metastatic LN.

## SLN mapping with ICTL

At least one SLN was identified in 38 dogs (97.4%) by at least two reviewers (Fig 5). The remaining dog had a SLN identified only by a single reviewer. Only lingual tumors required re-injection of contrast and a 12-minute scan. One SLN was identified in 13 dogs (33%), 2 were identified in 17 dogs (44%) of dogs, 3 were identified in 5 dogs (13%), and 4 SLNs were identified in 3 dogs (8%). ICC between reviewers on number of SLN was good (kappa = 0.81). The reviewers were in total agreement on the SLN(s) in 22 dogs (56%).

There was mild variability between reviewers in the inclusion of other additional nodes as SLNs. In 12 dogs (75%), an additional ipsilateral LN was deemed a SLN by at least one reviewer. Of these 12 dogs, an additional ipsilateral MLN was deemed an SLN by 1 (N = 4) or 2 (N = 2) reviewers, while in 5 dogs the ipsilateral MRLN was deemed an SLN l by 1 (N = 3) or 2 (N = 2) reviewers. In the last dog, one reviewer identified an additional ipsilateral MLN as an SLN while another identified the ipsilateral MRLN as an SLN.

In 2 dogs (13%) a contralateral MRLN was identified as a SLN by 1 (N = 1) or 2 (N = 1) reviewers. In 1 dog (6%) 1 reviewer identified an additional ipsilateral MLN as a SLN while another reviewer called an additional contralateral MLN a SLN. The most discordant results occurred in one dog in which one reviewer identified the right MRLN as an SLN, another reviewer identified the left MRLN as an SLN, and the third reviewer identified both MRLN as SLNs.

**SLN by tumor location.** The most common SLN(s) for mucosal lesions (N = 16 dogs) were the ipsilateral MLN (N = 11; 69%) or ipsilateral MLN + ipsilateral MRLN (N = 4; 25%) (Table 5). For unilateral oral tumors overlying/involving bone (N = 14), the most common SLN(s) were the ipsilateral MLN(s) (N = 13; 93%). For bilateral oral tumors overlying/

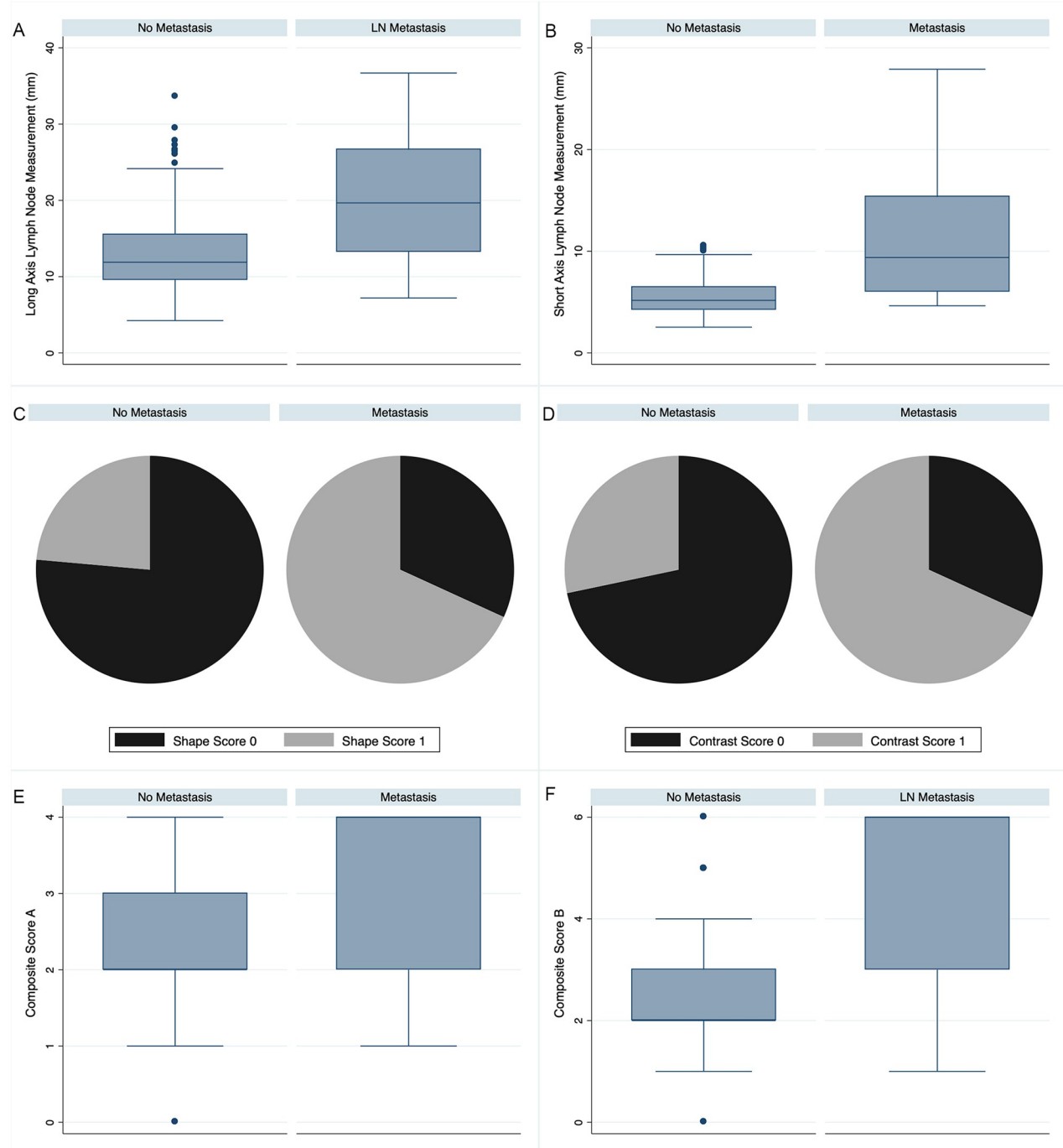

**Fig 3. Post-contrast computed tomography (CT) evaluation of cervical lymph nodes (LNs) in dogs with oral tumors.** Box and whisker plots demonstrate the median (center line) with the end of the boxes indicating the 25th and 75th percentiles and the whiskers indicating the adjacent values of values for post-contrast CT features for metastatic and non-metastatic LNs. Circles indicate outlier values, while pie charts are used for scores that only have two values. Features assessed for differentiation of metastatic from non-metastatic LNs include (A) LN long axis; (B) LN short axis; (C) LN shape score; (D) LN contrast pattern score; (E) Composite Score A; (F) Composite Score B.

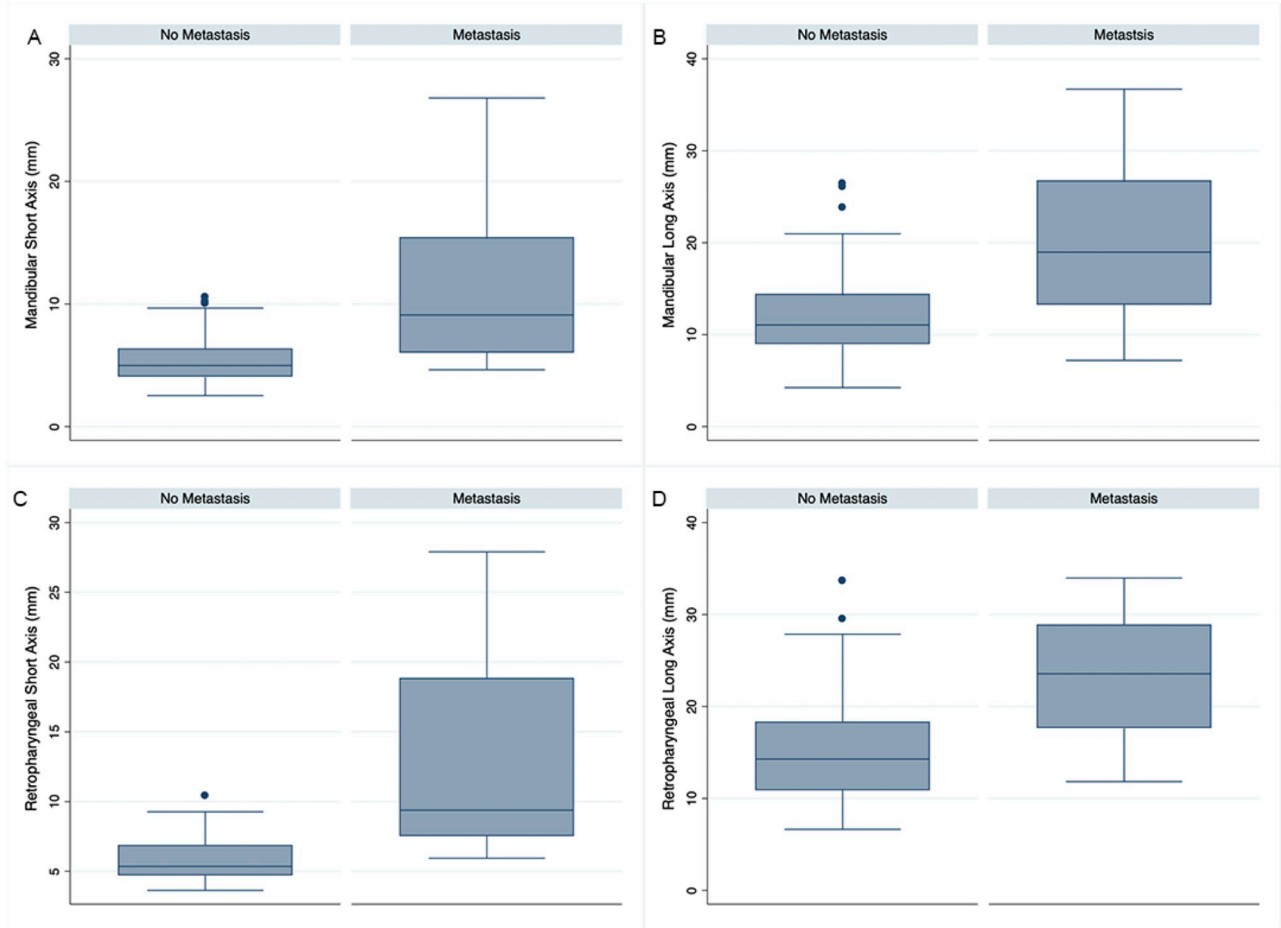

**Fig 4. Short and long axis measurements for cervical lymph nodes (LNs) in dogs with oral tumors.** Box and whisker plots demonstrating the median (center line) with the end of the boxes indicating the 25th and 75th percentiles and the whiskers indicating the adjacent values of post-contrast CT values for short and long axis measurements for metastatic and non-metastatic mandibular lymph nodes (A,B) and medial retropharyngeal lymph nodes (C,D). The circles represent outlier values.

involving bone (N = 5), 2 dogs had tumors that SLNs identified as the ipsilateral MLNs, while 3 dogs had SLNs identified bilaterally in the MLN. Lastly, for lingual tumors (N = 3), one dog had the ipsilateral MRLN only identified as the SLN and the remaining two dogs had bilateral MLN noted as SLNs.

**False negative rate of SLNB.** Of 234 LNs evaluated histopathologically, 22 metastatic LNs (9%) were identified in 13 dogs (33%). All 13 dogs with LN metastasis had at least 1 SLN identified. The only dog in this study that did not have an SLN identified did not have metastasis. When metastasis was identified in the cervical basin using histopathology (pN+ neck), the SLN was always metastatic. In 11 dogs with metastasis (85%), only the SLN(s) was metastatic, and in 2 dogs (15%) metastasis extended past the SLN to an additional MLN (i.e., lateral to medial, N = 2) and to the ipsilateral MRLN (N = 1). When multiple SLNs were identified in a case with metastasis (N = 9), 3 dogs (33%) had metastasis in both SLNs; the remaining 6 dogs (67%) had metastasis in only a portion of the SLNs. In cases where the SLNB was negative, no other nodes were identified as metastatic. Accordingly, the false negative rate of SLB was 0%.

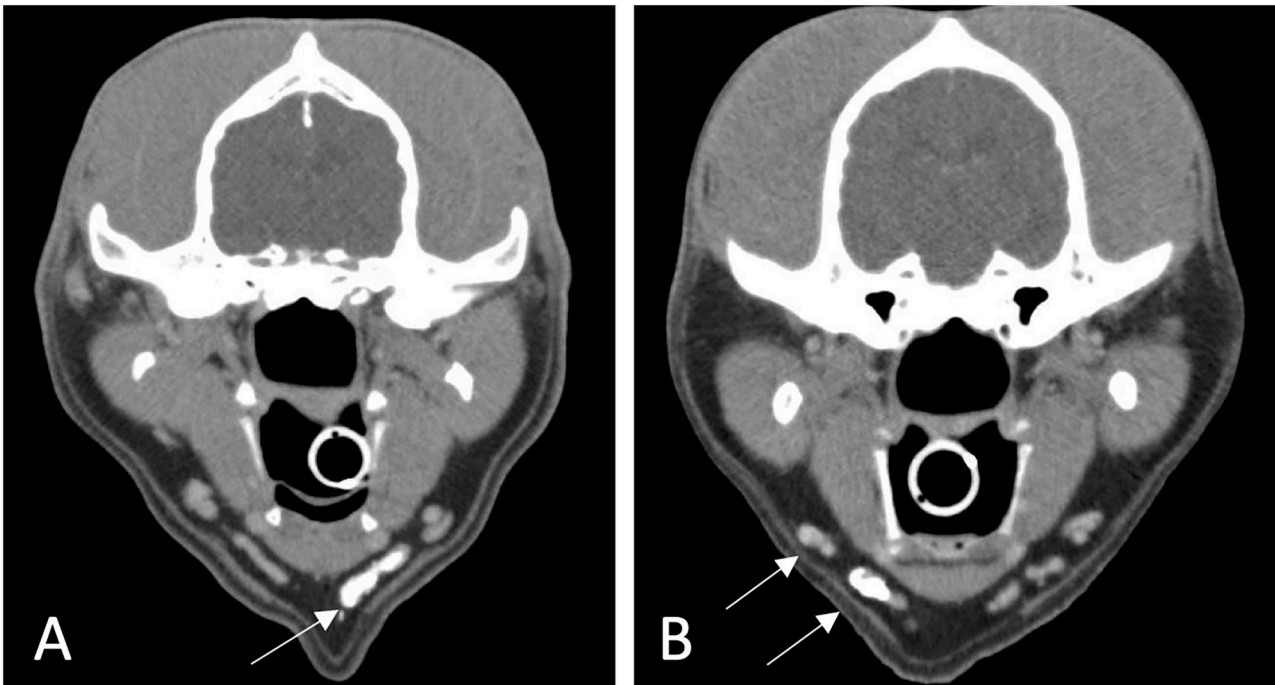

**Fig 5. Sentinel lymph node (SLN) detection on indirect computed tomography lymphangiography (ICTL).** Representative axial CT images (Soft tissue algorithm: W: 400 HU, L: 40 HU) demonstrating **A**: Concordance between all reviewers in identification of the left medial mandibular lymph node (MLN) as the SLN (arrow) and **B**: Discordance between reviewers, in which the medial MLN was the SLN (N = 2) or both the medial and lateral MLN (arrows) were identified as the SLNs (N = 1).

## ICTL evaluation

A total of 74 SLNs were identified in the 38 dogs in which at least 1 SLN was identified by two reviewers. ICTL features including contrast score and ICTL composite score were evaluated to compare metastatic and non-metastatic LNs, as well as SLNs and non-SLNs. For analysis of HU data only SLNs were included in evaluation. Evaluation was completed for the distinct LN groups (left and right medial MLN, lateral MLN and MRLN), and combined into MLN, MRLN and all LN.

When all LN or the MLN were evaluated as independent groups, ICTL features were not predictive of LN metastasis (Table 8 and Fig 6). When MRLN SLNs (N = 11) were evaluated as an independent group, both the central LN density and change in central LN density were significantly associated with LN metastasis (Table 8).

## Adverse effects associated with SLN mapping or END

No adverse effects consistent with anaphylaxis reactions occurred during the SLN mapping procedure. One dog with a lingual tumor developed tachycardia during injection, suggesting the procedure was painful. This was not observed during any other SLN mapping. Following cervical LN extirpation, the majority of dogs had a transient self-limiting post-surgical swelling. In 6 dogs (13%), swelling was sufficiently severe to warrant cervical ultrasound with fine needle aspiration cytology to confirm absence of postoperative infection. No further treatment was needed in four dogs and swelling resolved without intervention. One dog underwent marsupialization of a post-operative sialocele and a Penrose drain was placed while the dog was under anesthesia; had this dog not developed a sialocele, monitoring would have been

**Table 8. Significance of logistic regression model and receiver operator characteristic curve analysis for imaging features of the ICTL to predict lymph node (LN) metastasis in 69 identified sentinel lymph nodes in 38 dogs[a].**

| Characteristic | All LN (N = 234) | MLN (N = 156) | MRLN (N = 78) |
|---|---|---|---|
| ICTL contrast score | AUC: 0.52<br>p = 0.51 | AUC: 0.57<br>p = 0.2 | AUC: 0.56<br>p = 0.57 |
| ICTL Composite score | AUC: 0.58<br>p = 0.19 | AUC: 0.58<br>p = 0.2 | AUC: 0.64<br>p = 0.47 |
| **SLN Characteristic** | **All SLN (N = 69)** | **MLN (N = 58)** | **MRLN (N = 11)** |
| ICTL central LN density (HU) | AUC: 0.66<br>p = 0.08 | AUC: 0.63<br>p = 0.18 | AUC: 0.88<br>p = 0.04 |
| Change in central LN density versus postcontrast CT (HU) | AUC: 0.68<br>p = 0.19 | AUC: 0.62<br>p = 0.40 | AUC: 0.88<br>p = 0.04 |
| ICTL peripheral LN density (HU) | AUC: 0.61<br>p = 0.5 | AUC: 0.59<br>p = 0.71 | AUC: 0.67<br>P = 0.19 |
| Change in peripheral LN density versus postcontrast CT (HU) | AUC: 0.61<br>p = 0.65 | AUC: 0.59<br>p = 0.08 | AUC: 0.83<br>p = 0.06 |

LNs were analyzed grouped by all LN combined and with just the mandibular lymph nodes (MLN) or medial retropharyngeal lymph nodes (MRLN) grouped.
[a] one of the 39 dogs enrolled did not have an SLN identified thus there were no SLN features in one dog for inclusion.

performed. One dog (2.5%) had a culture confirmed infection at the lymphadenectomy site, requiring surgical explore and flush.

## Discussion

This is the largest study to date to predict the value of ICTL and SLNB for the cervical basin in dogs with oral tumors. At least one SLN was identified in 97% of dogs that underwent ICTL, and the SLNB false negative rate was 0%. These data demonstrate that ICTL with subsequent SLNB is an effective technique for pathologically staging canine oral tumors and may be an adequate substitute for END. Although, major side effects associated with END were rare and consistent with previous reports [10,20], transient post-surgical swelling should be expected, with some dogs (13% in our study) at risk for development of sufficiently severe swelling to prompt ultrasound and cytology to rule out postoperative infection. SLNB therefore offers an opportunity to minimize postoperative morbidity and adequately stage oral neoplasia while effectively cytoreducing metastatic disease. In human carcinomas, including those arising from oral and breast tissue, SLNB significantly reduces morbidity with similar 5-year survivals compared to END approaches [23–25,50]. In breast carcinoma, SLNB has replaced axillary dissection as the standard of care staging tool [50], thus minimizing hospitalization [51] and long-term pain and lymphedema in breast cancer patients [52]. This is particularly important in the era of COVID, given the need to reduce hospital visits and exposure of cancer patients to the virus [53]. This is also relevant to pet owners during this and future pandemics, as decreased postoperative complications in pet dogs will reduce the need for repeated veterinary visits that increase exposure risk to owners. Collectively, data from this study supports the need for additional studies to corroborate that SLNB in dogs with oral tumors is a rational standard approach to surgical staging and treatment. Like in human patients, as the focus is to reduce morbidity while maintaining or improving long-term quality of life and disease-free survival, the use of SLNB in pet dogs can be broadened to other tumor types with a moderate to high risk of LN metastasis.

There is no validated decision-making algorithm to accurately predict the true risk or incidence of cervical metastasis in dogs with oral tumors. For this reason, our institution routinely

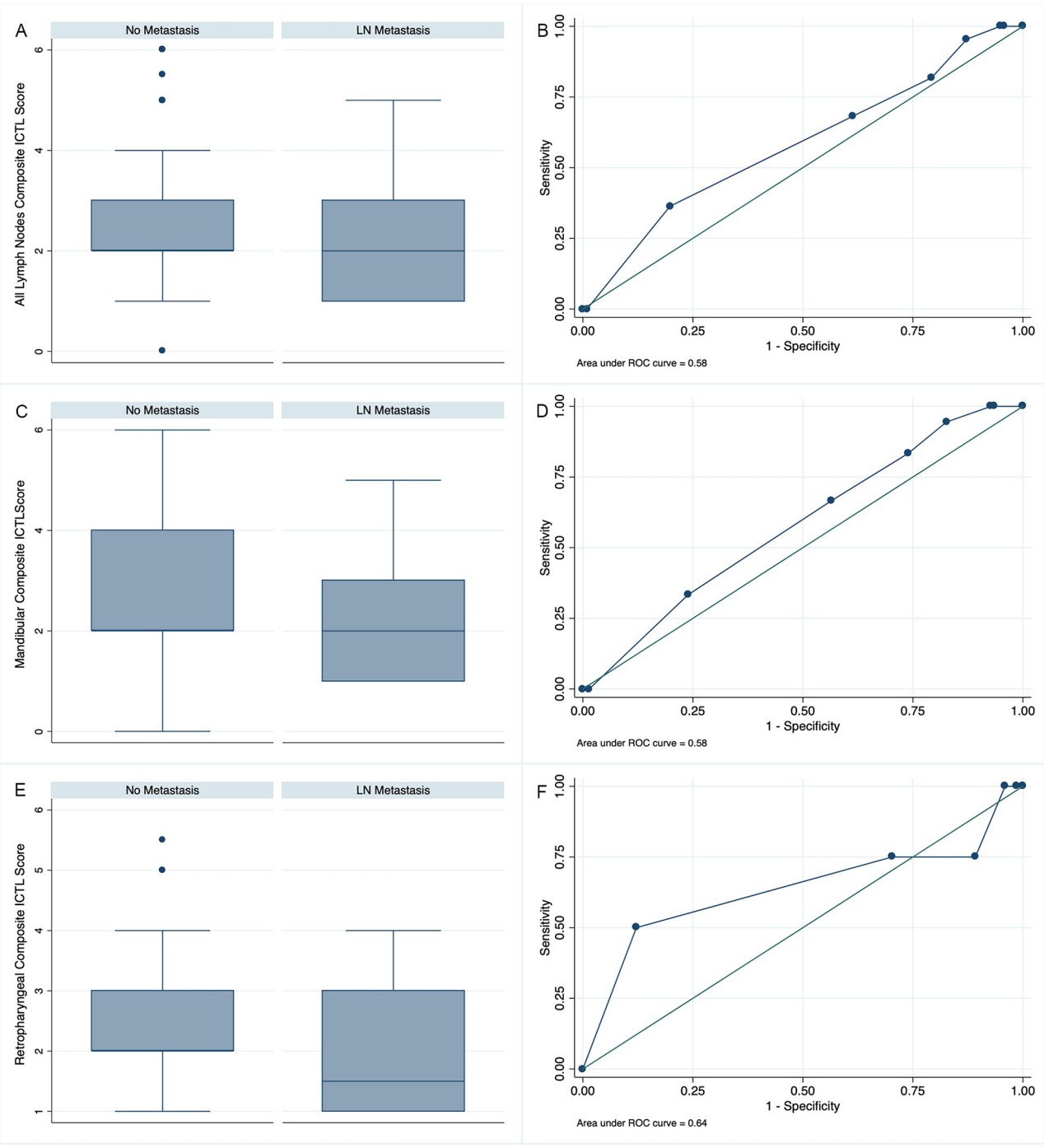

**Fig 6. Differences identified between composite indirect computed tomography lymphangiography (ICTL) scores between metastatic and non-metastatic cervical lymph nodes (LNs) in dogs with oral tumors.** Box and whisker plots demonstrating the median (center line) with the end of the boxes indicating the 25th and 75th percentiles and the whiskers indicating the adjacent values of differences in the composite ICTL score in non-metastatic LNs and metastatic LNs for all LNs combined (A) mandibular LNs (MLNs) alone (C), and medial retropharyngeal LNs (MRLNs) alone (E). The associated receiver operating characteristic curves demonstrate the predictive ability of the logistic regression model of the composite score for the detection of histopathologic metastasis for all LNs (B), MLNs alone (D), and the MRLNs alone (E).

performed END but sought to evaluate the ability of ICTL to identify the SLN to best predict the metastatic pattern. Of note, of the 39 dogs enrolled in this prospective study, only 3 dogs had metastatic extension past the identified SLN(s) on ICTL, and all additional metastatic LNs were ipsilateral to the identified SLN. This represents 7% of all dogs imaged, yet 23% of the dogs (N = 13) with LN metastasis. This highlights the potential importance of preoperatively identifying the metastatic status of the SLN, so that a transition to complete lymph node dissection can be considered. This finding mirrors human data, in which approximately 20% of patients with OSCC and cutaneous head and neck melanoma have LN metastasis beyond a positive SLN [29,54,55]. Accordingly, there is controversy surrounding the need for complete lymph node dissection when there is a positive SLNB. A future prospective study could evaluate the progression-free benefit of extirpating the remainder of the ipsilateral LN tract (e.g. all ipsilateral MLNs and RLNs) when a positive SLN is identified compared to only the SLN in dogs with oral tumors.

Because the knowledge of the LN metastatic status is vital to inform surgical considerations, our group, like others, have a strong interest in determining if specific CT and ICTL imaging features may predict metastasis in the cervical LNs and SLNs, respectively [31,32,39]. If specific imaging features are sufficiently accurate to predict metastasis, the need for LN sampling via aspiration or biopsy prior to definitive surgery may be negated. This has become increasingly important in some institutions or practices with limited on-site radiologists to perform ultrasound-guided sampling ability for deep-seated retropharyngeal LNs and/or to visually ensure that more than one MLN is sampled [56,57]. In this study, the post-contrast CT imaging features were superior to those of the ICTL, with most features demonstrating good accuracy at prediction of metastasis (Tables 7 and 8). However, there was significant overlap in the post-contrast CT and ICTL feature scores for metastatic and non-metastatic LN (Figs 3 and 6). This precluded the development of a clear clinical grading scheme to accurately identify LN metastasis. Of all features, regardless of when all LN were evaluated as a group, or the MLN or MRLN were evaluated independently, the short axis measurement was most clinically useful. All cases without metastasis measured < 10.5 mm. This finding mirrors a prior study that reported a width cut-off of 11 mm had high diagnostic accuracy in the prediction of cervical metastasis in dogs with OMM [13].

While we expected that ICTL features would provide better accuracy for prediction of metastasis compared to post-contrast CT features, the individual features (contrast score, measurement of HU) and the composite ICTL score had very poor accuracy. The ICTL central density was promising as a distinguishing feature for MRLN metastasis; however, confirmation of this finding requires a larger study. Interestingly, lower ICTL contrast and density scores were more likely to be associated with metastasis in this study. This may have occurred because metastasis prevents accumulation of contrast within the SLN, resulting in a decreased mean HU within metastatic LNs [32]. Our grading scheme could have been altered to give higher scores to LN with poor contrast enhancement. However, this would not have changed our results as lower ICTL contrast or density scores did not accurately predict metastasis.

Our data and results support that preoperative LN sampling is strongly recommended for canine oral tumors in the absence of END, due to the variability in drainage as previously described [14,18,19,31]. The most identified SLN(s), and metastatic LNs, were the ipsilateral MLNs. However, in almost 70% of dogs, multiple SLNs were identified using the ICTL technique. Yet only a subset of the SLNs were metastatic and our data could not reliably differentiate between SLNs that were metastatic or non-metastatic. We were unable to calculate sensitivity and specificity for the CT or ICTL scores because the determination of a clinically meaningful "cut-off" value for feature utility would have been arbitrary given the overlap between sample populations.

Our results compare similarly to previous efforts that used post-contrast CT with or without ICTL for SLN mapping and to evaluate SLNs for prediction of metastasis [12,13,32,38,39]. In one study of ICTL for detection of SLN metastasis in canine MCT and melanoma, ICTL alone was unable to predict histopathologic LN metastasis, although only 3 dogs in this study had oral or perioral tumors [39]. Conversely, in a study of canine mammary tumors, ICTL features including contrast pattern and post-contrast SLN density had high sensitivity, specificity and accuracy for prediction of metastasis, whereas SLN shape and size were less predictive [32]. Taking a different approach at SLN identification, one recent study combined preoperative lymphoscintigraphy with intraoperative gamma-probe and blue dye to investigate the cN0 neck in dogs with head and neck tumors [31]. Of the 8 oral tumors included, all dogs had at least one SLN identified preoperatively, and 4 of those LNs were histopathologically confirmed with metastasis [31]. Another study used indocyanine green near infrared fluorescence intraoperative imaging for SLN mapping with subsequent END to define its ability to predict metastasis in canine oral tumors [38]. While the metastatic LN was identified as an SLN on imaging, there was only one dog in the study with metastasis, somewhat limiting interpretation of its utility more broadly [38].

Of interest, there was variability in SLN determination and interpretation between reviewers. There is inherent variability in any reviewers' ability to accurately measure LNs. Although the differences were minimal and overall the ICC was excellent in our study with differences in measurements of only up to 1 mm. If utilizing a strict sub-mm size cut-off to determine metastasis, this variation could result in missing occult metastasis. Indeed, in our cohort, due to reviewer variability in interpretation, a metastatic deposit may have been missed in one dog. In our study, we included a radiation oncologist with two radiologists because of the unique reliance radiation oncologists have on CT imaging for radiation target delineation, such as cervical LN assessment in dogs with oral tumors. Moreover, as radiologists devote more effort towards non-academic teleradiology practice [56,57], it is vital to determine if new methods for image detection of metastatic LN may be done by other specialists accustomed to CT interpretation. In clinical practice, evaluation of the ICTL with multiple trained observers may be beneficial to ensure that all SLNs are sampled.

Caution must be applied as this data is in a relatively small patient population (N = 39) from a single institution with a standardized ICTL and SLNB procedure, and specialists accustomed to image interpretation similarly. Multi-institutional studies would hasten development of ICTL and SLNB approaches; however standardized imaging, CT positioning, data acquisition parameters, contrast use (dose, timing of imaging, number of injections), surgical approach, and histopathologic sectioning and evaluation would ideally be standardized. An additional limitation of our data set was that dogs in our study had a relatively low rate of LN metastasis, with 33% of dogs (9% of all extirpated LNs) in this study harboring metastasis. Dogs with a variety of tumor histologies and locations were included, which limited the power to evaluate the utility of SLNB for these subsets.

A final limitation to consider is that there is a risk that lymphatic occult metastasis was missed due to lack of scrutiny in evaluation of the SLNs. To minimize the risk that histopathology missed micrometastasis, we elected to perform serial sectioning. To date, there are only rudimentary recommendations for pathologic LN evaluation in oral neoplasia [14,58]. The subcapsular region is a common site for micrometastasis and serial sectioning optimizes evaluation of the LN periphery [58–60]. Thorough evaluation of the LN periphery also allows for detection of capsular invasion, which is prognostically significant [60]. We opted to section each LN entirely as serial sections perpendicular to the long axis. Compared to longitudinal bisection of lymph nodes ("bivalve" sectioning), serial sectioning perpendicular to the long axis allows for evaluation of greater circumference and, for enlarged nodes, greater surface

area. For example, in a lymph node measuring 2 x 1 x 0.5 cm, the bisected total surface area is 3.142 cm$^2$ and circumference of 9.688 cm. Serial sectioning of the same size node allows for evaluation of 2.146 cm$^2$ surface area (31.7% less) but 15.753 cm circumference (62.6% greater). In humans, it has been shown that serial step sections at 150μm intervals, used in combination with immunohistochemistry, can upstage nodal status in approximately 20% of patients [61,62]. Indeed, one study showed that routine pathologic evaluation only revealed metastasis in 13 patients (16%); however, incorporating serial step sections alone or with immunohistochemistry added 5 and 2 patients, respectively (totaling an additional 9% of patients), with LN micrometastasis [62].

## Conclusions

Canine oral tumors demonstrate variable LN metastasis but most often drain to a single MLN within the ipsilateral lymphocentrum. Post-contrast CT features such as short axis measurements may be useful for prediction of LN metastasis, with non-metastatic LN measuring less than 10.5 mm. SLN mapping with ICTL is a feasible diagnostic tool in canine oral tumors to guide accurate sampling for staging. Because no SLN imaging features could reliably distinguish metastatic from non-metastatic SLNs, cytologic or histopathologic SLN sampling is strongly recommended in the absence of END for accurate staging and treatment planning.

## Supporting information

**S1 Table. Clinical characteristics of the 39 dogs with oral tumors enrolled.**
(DOCX)

**S2 Table. Significance of logistic regression model and receiver operator characteristic (ROC) curve analysis for post-contrast and ICTL features of each mandibular lymph nodes (MLN) and medial retropharyngeal lymph node (MRLN) category separately from 39 dogs with oral tumors.** Significant values are shaded. When no data reported logistic regression failed.
(DOCX)

## Acknowledgments

The authors would also like to thank surgical oncologists, Pierre Amsellem for performing a portion of the END procedures, and Michelle Oblak for guidance on the ITCL SLN mapping protocol. Lastly the authors would like to thank Edward Mattan for modelling the calculation of LN area and circumference with serial sectioning.

## Author Contributions

**Conceptualization:** Stephanie Goldschmidt, Amber Wolf-Ringwall, Jessica Lawrence.

**Data curation:** Stephanie Goldschmidt, Amber Wolf-Ringwall, Jessica Lawrence.

**Formal analysis:** Stephanie Goldschmidt, Nikia Stewart, Christopher Ober, Cynthia Bell, Michael Kent, Jessica Lawrence.

**Funding acquisition:** Stephanie Goldschmidt, Nikia Stewart, Christopher Ober, Jessica Lawrence.

**Investigation:** Stephanie Goldschmidt.

**Methodology:** Stephanie Goldschmidt, Christopher Ober, Amber Wolf-Ringwall.

**Project administration:** Stephanie Goldschmidt.

**Writing – original draft:** Stephanie Goldschmidt, Jessica Lawrence.

**Writing – review & editing:** Stephanie Goldschmidt, Nikia Stewart, Christopher Ober, Cynthia Bell, Amber Wolf-Ringwall, Michael Kent, Jessica Lawrence.

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
