## [Decision Letter · Decision Letter 0]

18 Jan 2023

PONE-D-22-34386Contrast-enhanced and indirect computed tomography lymphangiography accurately identifies the cervical lymphocenter at risk for metastasis in pet dogs with spontaneously occurring oral neoplasia.PLOS ONE

Dear Dr. Goldschmidt,

Thank you for submitting your manuscript to PLOS ONE. After careful consideration, we feel that it has merit but does not fully meet PLOS ONE’s publication criteria as it currently stands. Therefore, we invite you to submit a revised version of the manuscript that addresses the points raised during the review process.

We look forward to receiving your revised manuscript.

Kind regards,

Sameh Attia, MS

Academic Editor

PLOS ONE

Journal Requirements:

"The authors thank the University of Minnesota Grant in Aid Program for funding this work (awarded to SG [PI], JL and CO)."

"University of Minnesota Grant in Aid

PI: SG, CO-I: JL,CO,NS

Grant #:  1801 - 11652 - 20562 - 4214572

https://research.umn.edu/funding-awards/grant-aid 

Reviewers' comments:

Reviewer's Responses to Questions

**Comments to the Author**

1. Is the manuscript technically sound, and do the data support the conclusions?

Reviewer #1: Yes

Reviewer #2: Yes

2. Has the statistical analysis been performed appropriately and rigorously? 

Reviewer #1: Yes

Reviewer #2: I Don't Know

3. Have the authors made all data underlying the findings in their manuscript fully available?

Reviewer #1: Yes

Reviewer #2: Yes

4. Is the manuscript presented in an intelligible fashion and written in standard English?

Reviewer #1: Yes

Reviewer #2: Yes

5. Review Comments to the Author

Reviewer #1: I read with great interest the Manuscript titled "Contrast-enhanced and indirect computed tomography lymphangiography accurately identifies the cervical lymphocenter at risk for metastasis in pet dogs with spontaneously occurring oral neoplasia." In my honest opinion, the topic is interesting enough to attract the readers’ attention.

-inclusion/exclusion criteria should be better clarified by extending their description.

-Discussions can be expanded and improved by citing relevant articles (I suggest authors to read and insert in references the following article PMID: 33400886)

Considered all this points, I think it could be of interest for the readers and, in my opinion, it deserves the priority to be published after minor revisions.

Reviewer #2: This well-written manuscript describes evaluation of indirect CT lymphangiography followed by sentinel lymph node biopsy to detect predict cervical lymph node metastasis, as compared to elective neck dissection. The methodology and results are well described and there are only a few areas where clarification is requested.

General comments:

There are several abbreviations that are used in the introduction that are only used infrequently in the remainder of the manuscript which occasionally confuse more than clarify. Recommend reassessing the use of FN, pN+, pN0 as these terms aren’t used that frequently in the manuscript. Also, in line 79, consider removing “(cN0)” as this is just a bit confusing when “cN0 neck” is used the remainder of the time.

Specific comments:

Page 4, line 91-94: Consider citing the actual NCCN guidelines here. Also, the NCCN guidelines have been updated since 2014 – please confirm that this statement is still accurate.

Page 6, lines 145-146: While the previous literature is references, consider confirming here that the injections were performed peritumorally in four quadrants (if that what was done) just to clarify that the number of sites is not the minor modification that was made to the previous published procedure.

Page 7, lines 166-167; Table 1; Fig 2: Please provide a bit more detail as to how the subjective score for metastasis was determined.

Page 11, Table 3: It’s a bit confusing why two dogs with APOT were included in this study. It’s clear that for one dog the original histo was SCC, but it’s not clear why the 2nd dog with APOT was enrolled. It’s clear from the published case report that CT lymphography was performed on this case (at the same time the CT of the thorax was performed), but it is unclear as to how it originally met the inclusion for this study. Please clarify. If this dog did not meet the enrollment criteria for this study, it should be removed from analysis.

Page 19, lines 362-363: pN+ is used infrequently in this manuscript; consider just stating, “…metastasis was identified in the cervical basin using histopathology,…”

Figures 2 & 3: Resolution is poor for both these figures – difficult to evaluate due to imagine quality.

6. PLOS authors have the option to publish the peer review history of their article (what does this mean?). If published, this will include your full peer review and any attached files.

Reviewer #1: **Yes: **Giorgio Bogani

Reviewer #2: No

---

## [Author Response · Author response to Decision Letter 0]

25 Jan 2023

I have uploaded a response to reviewers letter, below has the same information. 

The authors want to thank the editor and reviewers for their suggestions, please see specific responses below: 

Response to Editor

Response: The manuscript has been checked and the formatting has been updated where there were oversights. We have also correctly named the files and re-uploaded; we apologize for this oversight. 

2. Thank you for stating the following in the Acknowledgments Section of your manuscript: "The authors thank the University of Minnesota Grant in Aid Program for funding this work (awarded to SG [PI], JL and CO)." We note that you have provided funding information that is not currently declared in your Funding Statement. However, funding information should not appear in the Acknowledgments section or other areas of your manuscript. We will only publish funding information present in the Funding Statement section of the online submission form. 

"University of Minnesota Grant in Aid

PI: SG, CO-I: JL,CO,NS

Grant #: 1801 - 11652 - 20562 - 4214572

https://research.umn.edu/funding-awards/grant-aid

Response: The submitted Funding Statement is correct. We have removed the redundant wording from our Acknowledgements section; we apologize for this oversight in our initial version. No additional changes to the statement are required. 

Response: The authors have checked the references and are not aware of any retracted papers cited in our manuscript. EndNote20 automatically checks references for retractions and updates each time a library is opened. We are aware that no software is perfect for retraction watches. Please let us know if we have been unable to locate a retracted article flagged by the Journal. We have updated our reference list in our revised manuscript as we have addressed comments by the reviewers and have highlighted this in the cover letter. 

Response to Reviewer 1 

1. inclusion/exclusion criteria should be better clarified by extending their description.

Response: This has been updated 

2. Discussions can be expanded and improved by citing relevant articles (I suggest authors to read and insert in references the following article PMID: 33400886)

Considered all this points, I think it could be of interest for the readers and, in my opinion, it deserves the priority to be published after minor revisions.

Response: We appreciate the suggestions and have amended the discussion to include specific key points from the breast carcinoma literature and have applied this concept to our canine populations as well.

Response to Reviewer 2 

1. There are several abbreviations that are used in the introduction that are only used infrequently in the remainder of the manuscript which occasionally confuse more than clarify. Recommend reassessing the use of FN, pN+, pN0 as these terms aren’t used that frequently in the manuscript. Also, in line 79, consider removing “(cN0)” as this is just a bit confusing when “cN0 neck” is used the remainder of the time.

Response: Thank you for this suggestion, we have written out these abbreviations to improve clarity. We have elected to keep pN0 and pN+ in the body of the manuscript for completeness and to encourage this language to be utilized more frequently in veterinary medicine. Please advise, if you feel strongly about removing them all together. We have also added neck to areas where we use cN0 or cN+

2. Page 4, line 91-94: Consider citing the actual NCCN guidelines here. Also, the NCCN guidelines have been updated since 2014 – please confirm that this statement is still accurate.

Response: We appreciate the note that the NCCN guidelines are routinely updated. We have referenced the current NCCN guidelines for head and neck cancer and altered our wording to correctly reflect the NCCN recommendations. 

3. Page 6, lines 145-146: While the previous literature is references, consider confirming here that the injections were performed peritumorally in four quadrants (if that what was done) just to clarify that the number of sites is not the minor modification that was made to the previous published procedure.

Response: Thank you for noting this, we have updated to clarify that we also did 4-site peritumoral injections. 

4. Page 7, lines 166-167; Table 1; Fig 2: Please provide a bit more detail as to how the subjective score for metastasis was determined.

Response: This has been updated. 

5. Page 11, Table 3: It’s a bit confusing why two dogs with APOT were included in this study. It’s clear that for one dog the original histo was SCC, but it’s not clear why the 2nd dog with APOT was enrolled. It’s clear from the published case report that CT lymphography was performed on this case (at the same time the CT of the thorax was performed), but it is unclear as to how it originally met the inclusion for this study. Please clarify. If this dog did not meet the enrollment criteria for this study, it should be removed from analysis.

Response: This dog was included due to the biologically aggressive nature of the tumor and the presence of multifocal pulmonary metastasis. We have clarified the inclusion criteria in our methods to reflect our desire to include variable clinical scenarios by which dogs with biologically aggressive tumors were included.

6. Page 19, lines 362-363: pN+ is used infrequently in this manuscript; consider just stating, “…metastasis was identified in the cervical basin using histopathology,…”

Response: This has been modified slightly, we have still left in the pN+ as the comparative oncology, dentistry and oral surgery, and pathology groups would like to encourage the routine use of this terminology in veterinary medicine. Should the reviewer and editor want this to be removed in order to be published, we will defer and alter the wording.

7. Figures 2 & 3: Resolution is poor for both these figures – difficult to evaluate due to imagine quality.

Response: All Images have been updated to increase resolution, and the file names have been changed to ensure they meet PLOS ONE authorship guidelines

---

## [Decision Letter · Decision Letter 1]

16 Feb 2023

Contrast-enhanced and indirect computed tomography lymphangiography accurately identifies the cervical lymphocenter at risk for metastasis in pet dogs with spontaneously occurring oral neoplasia.

PONE-D-22-34386R1

Dear Dr. Goldschmidt,

We’re pleased to inform you that your manuscript has been judged scientifically suitable for publication and will be formally accepted for publication once it meets all outstanding technical requirements.

Kind regards,

Sameh Attia, MS

Academic Editor

PLOS ONE

Additional Editor Comments (optional):

Reviewers' comments:

Reviewer's Responses to Questions

**Comments to the Author**

1. If the authors have adequately addressed your comments raised in a previous round of review and you feel that this manuscript is now acceptable for publication, you may indicate that here to bypass the “Comments to the Author” section, enter your conflict of interest statement in the “Confidential to Editor” section, and submit your "Accept" recommendation.

Reviewer #2: All comments have been addressed

2. Is the manuscript technically sound, and do the data support the conclusions?

Reviewer #2: (No Response)

3. Has the statistical analysis been performed appropriately and rigorously? 

Reviewer #2: (No Response)

4. Have the authors made all data underlying the findings in their manuscript fully available?

Reviewer #2: (No Response)

5. Is the manuscript presented in an intelligible fashion and written in standard English?

Reviewer #2: (No Response)

6. Review Comments to the Author

Reviewer #2: (No Response)

7. PLOS authors have the option to publish the peer review history of their article (what does this mean?). If published, this will include your full peer review and any attached files.

Reviewer #2: No

---

## [Editor Report · Acceptance letter]

22 Feb 2023

PONE-D-22-34386R1 

Contrast-enhanced and indirect computed tomography lymphangiography accurately identifies the cervical lymphocenter at risk for metastasis in pet dogs with spontaneously occurring oral neoplasia. 

Dear Dr. Goldschmidt:

I'm pleased to inform you that your manuscript has been deemed suitable for publication in PLOS ONE. Congratulations! Your manuscript is now with our production department. 

Kind regards, 

on behalf of

Dr. Sameh Attia 

Academic Editor

PLOS ONE